# Hierarchical Machine Learning-Based Growth Prediction Model of *Panax ginseng* Sprouts in a Hydroponic Environment

**DOI:** 10.3390/plants12223867

**Published:** 2023-11-15

**Authors:** Tae Hyong Kim, Seunghoon Baek, Ki Hyun Kwon, Seung Eel Oh

**Affiliations:** 1Digital Factory Project Group, Korea Food Research Institute, Wanju-gun 55365, Jeollabuk-do, Republic of Korea; thkim@kfri.re.kr (T.H.K.); bseunghoon@kfri.re.kr (S.B.); kkh@kfri.re.kr (K.H.K.); 2Food Safety and Distribution Research Group, Korea Food Research Institute, Wanju-gun 55365, Jeollabuk-do, Republic of Korea

**Keywords:** plant factory, ginsenosides, machine learning, hydroponic cultivation, rottenness

## Abstract

Due to an increase in interest towards functional and health-related foods, *Panax ginseng* sprout has been in the spotlight since it contains a significant amount of saponins which have anti-cancer, -stress, and -diabetic effects. To increase the amount of production as well as decrease the cultivation period, sprouted ginseng is being studied to ascertain its optimal cultivation environment in hydroponics. Although there are studies on functional components, there is a lack of research on early disease prediction along with productivity improvement. In this study, the ginseng sprouts were cultivated in four different hydroponic conditions: control treatment, hydrogen-mineral treatment, Bioblock treatment, and highly concentrated nitrogen treatment. Physical properties were measured, and environmental data were acquired using sensors. Using three algorithms (artificial neural networks, support vector machines, random forest) for germination and rottenness classification, and leaf number and length of stem prediction models, we propose a hierarchical machine learning model that predicts the growth outcome of ginseng sprouts after a week. Based on the results, a regression model predicts the number of leaves and stem length during the growth process. The results of the classifier models showed an F1-score of germination classification of about 99% every week. The rottenness classification model showed an increase from an average of 83.5% to 98.9%. Predicted leaf numbers for week 1 showed an average nRMSE value of 0.27, which decreased by about 33% by week 3. The results for predicting stem length showed a higher performance compared to the regression model for predicting leaf number. These results showed that the proposed hierarchical machine learning algorithm can predict germination and rottenness in ginseng sprout using physical properties.

## 1. Introduction

Recently, consumers’ interest in health has increased. Functional and health-related foods are now preferred in their dietary life. Their interest in ginseng, one of the healthy functional foods or functional foods, is high. Ginseng (*Panax ginseng* C. A. Meyer) is one of the most important medicinal plants commercially cultivated in countries in Asia such as Korea, China, and Vietnam [1].

*Panax ginseng* is a perennial plant belonging to the Araliaceae family. It contains a significant amount of saponins, which have anticancer, anti-stress, neuroprotective, and anti-diabetic effects based on numerous physiological and pharmacological studies [2,3]. Generally, ginseng roots are harvested after growing 4 to 6 years via soil cultivation. They are used for medicinal prescriptions or health foods in the market due to the high concentrations of ginsenosides in roots. However, ginseng is exposed to the risk of soil-borne diseases caused by nematodes, fungi, and bacteria such as root rot and gray mold during such a long growing period [4]. In addition, there is a disadvantage of being unable to achieve consecutive cultivation. Moreover, available cultivation areas have been gradually decreasing recently [5].

Recently, numerous studies have shown that interest in consuming the entire part of sprout vegetables is increasing after growing ginseng seedlings using a sprout vegetable method. Sprout vegetables typically refer to vegetables that are germinated from seeds, leading to the development of cotyledons. They are used in their entirety at an earlier stage before full growth [6]. Plant seeds contain a significant amount of non-digestible components such as polysaccharides, tannins, and saponins. It has been reported that when these seeds are germinated to be grown as sprout vegetables, functional compounds are generated, leading to increases of beneficial constituents. In particular, ginseng sprouts (Ginseng, *Panax Ginseng* sprouts) have been receiving attention. They are recognized as a crop with high economic feasibility and potential value because the cultivation period is shorter than ginseng, about 3 to 6 weeks [7]. Additionally, it has been reported that the above-ground parts, such as leaves and stems, of one-year-old ginseng sprouts have higher ginsenosides concentrations than those of one-year-old ginseng roots [8]. At the same time, more than 30 types of ginsenosides have been reported in *Panax ginseng* sprouts [9].

As previously mentioned, the cultivation method using a plant factory with a soil-less cultivation method has a short cultivation period for ginseng sprouts [4]. Among the various methods, the use of hydroponics has been reported to offer advantages in terms of minimizing the influence of external conditions and controlling internal environmental factors such as light, temperature, humidity, and CO_2_ concentration within plant growing facilities, leading to high yields and improved quality through commercialization [10]. Studies have reported optimal conditions for cultivating ginseng sprouts based on a closed-greenhouse plant production system utilizing the hydroponic-cultured method (short-term hydroponic-cultured ginseng) [11]. Studies have also reported that extraction yields of saponins among ginsenosides in ginseng sprouts are affected by external environments such as temperature, nutrient solvent, and liquid-to-solid ratio [12]. Additionally, studies on LEDs, one of the external control environment variables, have reported that the cultivation period and the number of plants per unit area are increased when the light source, an essential item for crop growth, is configured in a high-growth environment condition. In particular, LEDs can serve as compact light sources to facilitate space utilization, allowing control over light intensity and quality through specific wavelength combinations [13]. When the red-light ratio is high, concentrations of ginsenoside components are increased due to the impact of far-red light on leaf expansion [14]. Furthermore, environmental factors such as air temperature, photosynthetic photon flux density, and the electroconductivity (EC) of the nutrient solution within a closed greenhouse using the hydroponic-cultured method can influence growth outcomes [15,16].

Although various studies have been conducted on the growth environment of hydroponic plant production systems, there is no standardized model for obtaining large quantities within a short cultivation period [17]. To overcome these problems, studies have been conducted to analyze the contents of ginsenosides (Rg1, Rb1, Rd, etc.) in ginseng sprouts in various growing environments [18]. However, most of these studies analyzed improvements in the content of ginseng sprouts’ indicator substances according to changes in various environmental conditions in a short-term hydroponic-cultured environment, whereas studies that predicted physiological characteristics of ginseng sprouts according to environmental changes were insufficient [15]. Recently, studies on growth process analysis and disease discrimination of plants or ginseng sprouts using closed plant factory systems are gradually increasing [19,20]. Research is being conducted to improve efficiency and productivity through controlling the growing environment by applying information and communication technologies (ICTs) when growing ginseng sprouts in a hydroponic-cultured environment [18]. For example, one study has recently reported that, according to the wavelength range of LED, an important environmental condition in hydroponic cultivation, the effect of inhibiting plant pathogenic fungi and the growth of leaves and rhizomes are promoted [21,22]. However, it has been reported that, when chemical nutrition is used to promote the growth of ginseng sprouts in a closed hydroponic environment, there is a high possibility that the decay rate of ginseng sprouts will increase due to environmental changes such as nutrient solution pH concentration. In addition, the occurrence of algae may increase due to LEDs. If proper water replacement is not performed, roots may rot due to lack of oxygen in the water. Additionally, the root weights of ginseng sprouts are higher at an air temperature of 20 °C than that at 25 °C, and the plant growth density is higher when the electrical conductivity is higher. If an appropriate environment is not maintained, an environment with a high density due to rapid spread of decay with a hydroponic-cultured method will require the disposal of crops due to problems arising in that specific environment. Furthermore, there have been no reported studies on predicting the early stages of disease occurrence or quality in ginseng sprouts for the purpose of enhancing productivity.

Therefore, in this study, various growth environment conditions were created based on the hydroponic cultivation method for ginseng sprouts. Data were obtained for the growth process and environment using sensors and direct measurement methods. Based on acquired data, a hierarchical machine learning model was proposed to predict the rottenness and germination of ginseng sprouts and leaf numbers and stem length one week after the data was acquired. This is considered to be able to improve the productivity of ginsenoside extracted from ginseng sprouts by determining the growth process of ginseng sprouts in advance for a hydroponic-cultured method to be used in the future.

## 2. Results

In this study, a hierarchical machine learning model was proposed to classify ginseng sprouts’ germination and rottenness from 0 to 3 weeks based on sensor data obtained from a hydroponic-cultured environment and growth information measured during the cultivation process. Classification results for the germination and rottenness of the ginseng sprout samples were then utilized as additional input features to predict leaf number and stem length of ginseng sprouts as the second step in the proposed machine learning model.

### 2.1. Classification Result for Germination and Rottenness of Panax Ginseng

Germination prediction, classification accuracy, precision, recall, and F1-score for week 1, week 2, and week 3 of ginseng sprouts grown in a hydroponic-cultured environment using sensor and hand-measured data from week 0, week 1, and week 2 are shown in Table 1, respectively.

As a result of classifying germination 1 week after data acquisition using the average growth environment data in Week 0, an average accuracy of 99.13 and an average F1-score of 99.42 are shown. Additionally, the germination classification results for Week 2 and Week 3 showed a high F1-score of approximately 99%. Figure 1 is a representative confusion matrix of the germination classification algorithm. The diagonal confusion matrix in Figure 1 shows the number of correctly classified sample where the real germination sample has a germination classification of 174 for Week 1. The number of samples falsely classified as no germination was 1.6 for Week 1. As a result of the germination classification, a hierarchical machine learning model, the classification result is a binary result where the sample will be germinated or not germinated after 1 week from the data acquisition state. About 230 testing samples were randomly selected from the total data samples; in the case of Week 3, the average number of predicted germinated ginseng was about 202.6 among 202.8.

The performance of the machine learning classification algorithms in classifying rottenness status one week ahead during the growth process of ginseng sprouts is presented in Table 2. From Week 1 to Week 3, performance indicators including accuracy, precision, recall, and F1-score increased from an average of 83.5% to 98.9%. In Week 1, the average accuracy was approximately 98.8%, showing a high value, while the recall was relatively lower at around 81.5%.

Additionally, as shown in Figure 2, in Week 1, most of the validation ginseng sprouts did not decay. On average, out of 10 decayed ginseng sprout samples, approximately 8.6 were classified as rotten. In Week 3, the final average precision of the rottenness classifier model increased from 87.2% to 99% and the recall increased from 81.5% to 98.8%. In addition, the number of misclassified rotten ginseng samples decreased as compared to Week 1 and 2.

### 2.2. Regression Result for Predicting the Number of Leaf and Length of Stem for Panax Ginseng

Results of the regression model for predicting the number of leaves of ginseng sprouts in non-decayed ginseng sprouts according to the classification results of rottenness and germination of ginseng sprouts grown in a hydroponic-cultured environment are shown in Table 3.

After planting ginseng sprouts in the hydroponic bed and using environmental and growth data, predicted leaf numbers for Week 1 exhibited average values for nRMSE, MAE, and R of 0.27, 0.79, and 0.96, respectively. For Week 3, the nRMSE decreased by approximately 33%, while changes in MAE and R values were negligible. As shown in Figure 3, actual leaf number values from Week 1 to Week 3 and predicted leaf number values from the regression model exhibited a high correlation with an R value of approximately 0.98.

Table 4 shows the results for predicting the stem lengths of ginseng sprouts. For Week 1, the nRMSE, MAE, and R values were notably high, at 0.006, 0.014, and 0.99, respectively. In contrast, for Week 3, the nRMSE and MAE values were slightly higher, at 0.02 and 0.04, respectively, than those for Week 1. These prediction results for ginseng sprout stem length showed a better performance compared to the results of the regression model for predicting the leaf number of ginseng sprouts. In the case of ginseng sprout stem length, it tended to increase linearly as the growth period became longer. However, the leaf number did not show a linear increase, leading to lower prediction results.

As shown in Figure 4, there was almost no error between measured stem lengths of ginseng sprouts and predicted stem length values.

## 3. Discussion

In this study, various environmental conditions were constructed within a hydroponic-cultured environment. Data about the growth process and growth environment of ginseng sprouts were then obtained based on sensors and actual measurements. Based on acquired data, a hierarchical model utilizing machine learning algorithms was proposed to classify and predict growth results of ginseng sprouts according to growth period. To achieve this, three supervised machine learning algorithms (ANN, RF, SVM) and optimization techniques were applied to enhance the classification and prediction outcomes.

In this study, the hydroponic system’s conditions for ginseng sprout cultivation were controlled by altering factors such as LED lighting, temperature, pH, EC, and nutrient solution composition. According to a recent study, the type of water used in the nutrient solution preparation process plays a significant role among factors determining plant growth in hydroponic-cultured systems. For lettuce that can be grown in a hydroponic-culture environment, the same one used for growing ginseng sprouts, a study has analyzed the growth and chlorophyll content of lettuce using different types of water sources for the nutrient solution, including tap water, bottled water, and groundwater. It found that groundwater, which contained various dissolved minerals, promoted plant growth by supplying nutrients to the plant during growth [23]. Therefore, as a further study, conducting studies to enhance the growth of ginseng sprouts and enrichment of functional compounds by incorporating different types of water sources into the nutrient solution during the manufacturing process could be pursued. This approach may allow for advanced prediction of ginseng sprout germination or rottenness under various environmental conditions.

Recently, there has been a growing trend in the field of agriculture to apply artificial intelligence or deep learning techniques to tasks such as assessing the shape and variety of specific crops, diagnosing diseases, or categorizing agricultural products, and predicting harvest yields [24]. For instance, the research by Jayapal et al. [25] utilized image processing and deep learning to predict the occurrence of root rot disease of ginseng roots in advance. Additionally, 5516 ginseng image datasets were constructed, and a deep learning model based on the convolutional neural network algorithm was developed to automatically classify ginseng grades. As a result, the accuracy of classification of ginseng was about 94.5%. Results were similar to those of experts. However, since there is no study applying a deep learning algorithm based on the environment and ginseng sprout growth results obtained in a hydroponic-cultured environment to predict the quality of ginseng sprouts, a direct comparison with the results of this study is difficult. In the present study, the average accuracy of the AI algorithm was 99% for the germination rate and 92.3% for the rottenness rate of ginseng sprouts. In the case of the rottenness rate classification model, as the growth period increased, the number of decayed ginseng sprouts also increased. Upon analyzing the classification model results for Week 1, it was observed that the amount of rotten ginseng sprout data used during the learning process was lower than the amount of data for non-rotten sprouts. In the future, performance can be improved by either obtaining a more balanced amount of data for decayed and non-decayed ginseng sprouts or by adding a correction process for unbalanced learning data.

## 4. Materials and Methods

### 4.1. Data Sample

#### 4.1.1. Sample Material

In this study, ginseng sprouts harvested in Hampyeong-gun, Jeollanam-do, Republic of Korea (cheonjiseunghyeon-farm) were purchased and used as samples (Figure 5). They were first washed with flowing water to remove soil on sprouts. They were then immediately transplanted to the test bed. For each treatment, 126 1-year-old roots and 63 2-year-old roots were transplanted. The experiment was repeated three times for about 21 days.

#### 4.1.2. Configuration of Growth Environment

This growth experiment was performed after configuring a hydroponic-cultured test bed system as shown in Figure 6. Ginseng sprouts in the treatment group were grown with a deep flow technique (DFT) by filling a bed of 515 mm × 400 mm × 180 mm with 20 ℓ of water.

One-year-old and two-year-old ginseng seeds were transplanted into sponge beds and an initial 72 h shaded environment was created to facilitate germination. Humidification was provided every 30 min to prevent ginseng seedlings’ heads from drying out. In each treatment group, fine air bubbles were introduced for ginseng cultivation using a 25 mm bubble stone, ensuring a continuous and smooth supply of oxygen to sprouts for 24 h. For plant growth illumination, a light-emitting diode (LED) and 4 rows of 1118 mm long bar-type LEDs (Plant Grow Lighting, Custom-made Spectrum, ALLIX Co., Jeonju, Jeollabuk-do, Republic of Korea) were applied and provided light for 9 h (9 a.m. to 18 p.m.) for plant growth. To prevent the growing environment temperature from rising due to LED lightening for plant growth, fans (small CPU cooling fans, 120 mm) were installed on the left and right sides to discharge heat to the outside.

Four different treatment groups were established for cultivating ginseng sprouts. These groups included a control treatment (CT) using a commercially available standard nutrient solution, a hydrogen-mineral treatment (HMT) involving a hydrogen-mineral nutrient solution, a Bioblock treatment (BT) utilizing only bio blocks within the nutrient solution, and a highly nitrogen concentrated treatment (NCT). The control treatment’s (CT) standard nutrient solution (manufacturer: Natural Alkali 9.2, KSN Bio, Gwangju, Gyeonggi-do, Republic of Korea) was prepared by mixing the nutrient solution and water in a ratio of 100:1. The hydrogen-mineral nutrient solution had a pH of 5.13 and an EC (ds/m) of 0.31. Its composition (%) included K 9.57, Ca 8.09, Mg 14.17, Na 17.61, Cl 41.15, NO_3_-N 0.61, S 4.82, P 2.24, Zn 1.31, Cu 0.05, and Mn 0.95. The highly concentrated nitrogen nutrient solution (Peptide, SaeWon Life Science, Seoul, Republic of Korea) had a pH of 6.53 and an EC (ds/m) of 0.12. Its composition (%) included K 12.62, Ca 18.08, Mg 4.30, Na 9.44, Cl 46.08, and NO_3_-N 4.04. For each treatment group, the water in the hydroponic cultured environment was replaced weekly. To prevent water spoilage due to evaporation from temperature and nutrient solution absorption by ginseng sprouts, the nutrient solution was remixed and replaced every three days [26]. Bioblocks (manufacturer: Soulbio (Soulbio, Daejeon, Republic of Korea), radiant energy 3.57 × 102 W/m^2^ μm, 37 °C, emissivity 5–20 μm standards 0.926) were used. They were manufactured by dissolving material crushed into particles of 300–1000 mesh and solidifying at a temperature of 1500–2000 °C which had a weight of approximately 300 g, a diameter of 850 mm, and a thickness of 23.5 mm.

### 4.2. Data Processing

#### 4.2.1. Data Sampling

The method for measuring growth results of ginseng sprouts in the process of growing ginseng sprouts in a hydroponic environment is described as follows. After transplanting ginseng sprouts into beds as shown in Figure 7, measurements were taken at 7-day intervals over a span of 21 days, including a 72 h blackout period, to assess changes in stem length, root length, rhizome, leaf numbers, germination, and the existence of rottenness. Growth measurements were obtained by referencing the Agricultural Science and Technology Research Analysis Criteria for data acquisition [27]. Total length of ginseng sprouts, including seedlings’ heads and stem length, was measured from seedlings’ heads to the maximum height of leaves. Root length was measured using a ruler from seedlings’ heads to the lowest point of roots. The rhizome was measured using a Metric 500-704-20 waterproof IP67 digital vernier caliper (Mitutoyo Co., Kawasaki, Japan) from a point 1 cm below the seedlings’ heads. Each sample was measured individually and then transplanted back to its original position to obtain growth variation data over the cultivation period. The leaf number was measured for all leaves of a single ginseng sprout. Germination status was determined by assessing whether cotyledons were fully unfolded and by observing the presence of germination from seedlings’ heads throughout the growth period. Criteria for determining decay were established by referencing agricultural science and technology research encompassing factors such as gray mold disease, stem withering disease, root decay, and others [27].

Various sensors were used to obtain information about the growth of ginseng sprouts regarding the hydroponic-cultured environment. As shown in Figure 7, ginseng sprout growth-environment parameters were air temperature, humidity, water temperature, pH, electro conductivity, dissolved oxygen. These environmental conditions were measured using a temperature and humidity sensor (DHT11, SEN030000, YwRobot, 0~50 °C ± 2 °C, 20~90%RH ± 5%), a water temperature sensor (DS18B20, SEN050007, −10~85 °C ± 0.5 °C), a pH meter (SEN0161, DFROBOT, 0~14 pH ± 0.1 pH (25 °C)), an analog electrical conductivity meter (SKU:DFR0300, DFROBOT, 1~20 ms/cm ± 0.1 ms/cm), and a dissolved oxygen sensor (SKU:SEN0237, DFROBOT, 0~20 mg/L) [16,28]. All sensor data were collected using Arduino Mega 2560 R3 (SZH-EK028, Arduino, MA, USA) connected with a workstation though a c-type USB with a baud rate of 9600.

Figure 8 shows an example of the growth process of ginseng sprouts in a hydroponic-cultured environment for 21 days. As the growth period increased, as shown in Figure 8, germination of ginseng sprouts and the number of leaves could be identified (white box).

#### 4.2.2. Data Preprocessing

In this study, collected data during growth of ginseng sprouts came in two data formats: time-series data and discrete data. Among environment condition parameters, air temperature, water temperature, humidity, pH, EC, and DO data were time-series data. They were sampled and transmitted to workstation every minute for each week. Received sensor-based growth environment condition parameters (temperature, water temperature, humidity, pH, EC, DO) were analog signal. These time-series data were converted into discrete data by calculating the average, maximum, and minimum of each parameter which will be implemented as input feature vector for machine learning algorithms. In addition, converting time-series data into discrete feature-type data increased the number of features from 7 to 18 for classifying and predicting the output of growth ginseng sprouts.

#### 4.2.3. Input and Output Feature Formation for Machine Learning Algorithms

Next, to develop a machine learning model to classify the rottenness and germination status of ginseng sprouts, and predict the leaf number and stem length based on the classification results, input and output matrices were constructed (Table 5). Growth information measured during ginseng sprout’s growth process and pre-processed cultivation environment sensor data were combined to form input feature variables. The initial size of variables used as input for the 1st step of hierarchical machine learning algorithm was 24 for each sample. Among 24 input features, 18 features were calculated from time-series sensor data transformed into discrete type and the remaining 6 features were hand measured data about ginseng sprout growth and the cultivation environment. The growth environment established for cultivating ginseng sprouts was categorized into treatment groups, ginseng age, and cultivation periods. Types of treatment groups, age of ginseng, and periods were parameterized in order to use them as input features. As an example, the labeling for the standard treatment group was parameterized as 1, the hydrogen-mineral treatment (HMT) group as 2, the Bioblock treatment (BT) as 3, and the highly concentrated nitrogen treatment (NCT) as 4, to construct the input matrix for a hierarchical machine learning model. Additionally, for cultivation periods, variable values were labeled as 1 for 0 weeks, 2 for 7 days, 3 for 14 days, and 4 for 21 days. Measured values from the growth process of ginseng sprouts and the growth environment sensor data were directly used as input variables without undergoing parameterization. Next, for the rottenness and germination status of ginseng sprouts, if a sample of ginseng sprouts decayed during that week, the corresponding result was labeled as 1. If it did not decay, the result was labeled as 2. Additionally, if germination occurred, the result was labeled as 1. If germination did not occur, the result was labeled as 2. Collected growth data and environment data were combined into one matrix for each week.

### 4.3. The Process of Hierarchical Machine Learning Algorithm for Classifying and Predicting Growth of Ginseng

In this study, a hierarchical model was proposed to classify and predict the state of ginseng sprouts during their growth process in a hydroponic-cultured environment using three machine learning algorithms. As shown in Figure 9, growth environment sensor data and growth process measurement data of ginseng sprouts were acquired. Pre-processing of acquired raw data was then performed using a moving average filter to remove noise from sensor data. After that, data were transformed into discrete type as mentioned above and applied as input feature for machine learning algorithms after going through a data labeling process. To verify the performance of the ginseng sprouts classification and regression model, a cross-validation technique was applied with the k value set to 5. The number of data samples acquired from the experiment was 1148, of which 918 (80%) samples were randomly extracted and learned. The remaining 230 (20%) samples were used as model verification data. A total of five repetitions were performed.

As shown in Figure 10, a detailed machine learning model was proposed based on the growth environment and growth process measurement data. In the first step, the rottenness and germination status of ginseng sprouts were classified. The combination of classification results appeared in four categories based on classification algorithm: (1) ginseng sprouts germinated without decaying; (2) sprouts germinated with decaying; (3) sprouts did not germinate or become rotten; and (4) sprouts did not germinate but decayed. According to these four combinations, in cases where ginseng sprouts experienced decay, data for leaf number and stem length of those samples were not used for the subsequent regression model in the next step. However, if ginseng sprouts did not experience decay, data were applied to the prediction model in the following step. For instance, if the result of rottenness and germination classification results are 1 and 2, respectively, the specific data sample will be returned.

#### 4.3.1. 1st Stage of Germination and Rottenness Classification

In this study, two classification models were developed for the first stage of the proposed model. The first one pertained to the rottenness status of ginseng sprouts and the second one pertained to whether ginseng sprouts germinated. Both models utilized three supervised learning-based classifiers: (1) artificial neural network (ANN); (2) support vector machine (SVM); and (3) random forest (RF) (Figure 11). ANN is a layered network-based mathematical model that maps and finds relationships between input and output variables. It is suitable for non-linear relationships between data samples. The structure of the ANN in this study comprised one input layer, three hidden layers with an initial hidden node of 150 for each layer, and an output layer. For the classification model, the output layer was sequenced with softmax and a classification layer [29].

SVM is a method to find boundaries between each class of ginseng sprout rottenness and germination in form of a hyperplane. It minimizes the structural risk by forming nonlinear kernel by nonlinear decision. It is therefore important to find optimum boundaries or hyper-planes to accurately classify [30]. RF is a one of the most commonly used ensemble machine learning classifiers that can create a tree-structured classifier where the random vector is independent identically distributed and each tree casts a unit vote for the most popular class at the input variable [31,32]. By comparatively analyzing the F1-score, one of the performances of three supervised machine learning classifiers, the training model and results of the model with the highest F1-score were derived (Figure 11).

#### 4.3.2. Prediction of Number of Leaves and Length of Stem

The following 2nd step of proposed hierarchical model consists of a regression model that predicts leaf number and stem length of ginseng sprouts by adding classification results in the first hybrid classification in input feature of prediction model. In the case of the regression model, it is composed of the same flow chart as the classification model except the size of input feature as well as the final layer of each machine learning algorithm is replaced with a regression layer instead of softmax and a classification layer. Structure and parameters of the machine learning algorithms were applied in the same way.

### 4.4. Optimization

Optimization techniques were applied to enhance the performance of both the classifier for categorizing ginseng sprouts’ germination and rottenness status and the regression model for predicting the leaf number and stem length of sprouts. The global gray wolf optimization (GWO) algorithm, which could yield faster results than the commonly used genetic algorithm or Bayesian optimization algorithm, was applied. GWO is a meta-heuristic-based optimization technique derived from the hunting method of a pack of gray wolves. It was developed based on a model in which wolves move with a hierarchical structure to find the optimal target value. This meta-heuristic optimization technique can solve complex combinatorial problems. It has advantages such as simple originality, clear optimization fixation, and flexibility in changing the cost function [33]. In this study, there are two cost functions due to the hierarchical composition of two models: a classifier for categorizing ginseng sprout growth results and a regression model for predicting growth results. The first one is composed of a cost function (Formula (1)) that minimizes the 1-F1-score of the classifier. The second one is a cost function that minimizes the nRMSE for the regression model (Formula (2)). To enhance performance, the hyper-parameters of the three machine learning algorithms (ANN, RF, SVM) applied in this study were set as design parameters:Loss_1_ = 1 − min(F1-score/100)(1)
Loss_2_ = min(nRMSE of number of leaf prediction model)(2)

### 4.5. Performance Evaluation

In this study, a classifier that classifies whether ginseng sprouts are decayed or/and germinated and a regression model that predicts the leaf number and stem length during the growth process of ginseng sprouts were used. To evaluate the performance of the classifier and the regression model, accuracy, precision, recall, and F1-score were calculated for the classifier as shown in the formula below (Formulas (3)–(6)). For the classifier, the predicted value obtained from the machine learning algorithms was compared with the actual value to construct a confusion matrix, which was then utilized to calculate the four performance indicators mentioned above. Next, for the regression model, nRMSE, MAE, and R values were extracted to evaluate and analyze the performance of the regression model (Formulas (7)–(9)) [34].
Accuracy = (# of correctly classified classes/total # of classes) × 100(3)
Precision = True positive/(True positive + false positive) × 100(4)
Recall = True positive/(True positive + false negative) × 100(5)
F1 score = 2 × (precision × recall)/(precision + recall) × 100(6)
(7)$$nRMSE=\frac{\sqrt{\frac{\sum_{i=1}^{N}{\left(y_{i}-\hat{y_{i}}\right)}^{2}}{N}}}{\bar{y}}$$
(8)$$MAE=\frac{1}{n}\sum_{i=1}^{n}\left|y_{i}-{\hat{y}}_{i}\right|$$
(9)$$R=\frac{\sum \left(x_{i}-\bar{x})(y_{i}-\bar{y}\right)}{\sqrt{\sum {\left(x_{i}-\bar{x}\right)}^{2}\sum {\left(y_{i}-\bar{y}\right)}^{2}}}$$

All data processing and machine learning algorithm were performed using MATLAB program (R2022b, MathWorks, Natick, MA, USA). The supervised machine learning algorithm was trained and tested based on a windows system with Intel^®^ Core™ i7-12700@ 3.70 GHz and NVIDIA GeForce A4000 with 32 GB graphic card.

## 5. Conclusions

Our proposed hierarchical machine learning-based method allowed us to obtain accurate growth results prior to the actual growth of a ginseng sprout. This model can reduce the cost and time of growing ginseng sprouts, as well as determine the rottenness of ginseng sprouts, allowing us to remove the contaminated sample prior to it spreading to nearby ginseng sprouts. There were several issues that increased the experimental complexity and time consumption while collecting sensor data as well as hand measurement data. Furthermore, there was the possibility of contamination during the process of removing ginseng sprouts from growth trays for measurement and then repositioning them. Therefore, for further research, a system could be developed using cameras to capture the growth process and algorithms to automatically extract growth results such as leaf number and stem length from acquired images. This approach could allow for an automated derivation of ginseng sprout growth results in a hydroponic environment [34]. In future, our proposed model can be applied to harvests in fully automated hydroponic plant factories by predicting growth performance.

## Figures and Tables

**Figure 1 plants-12-03867-f001:**
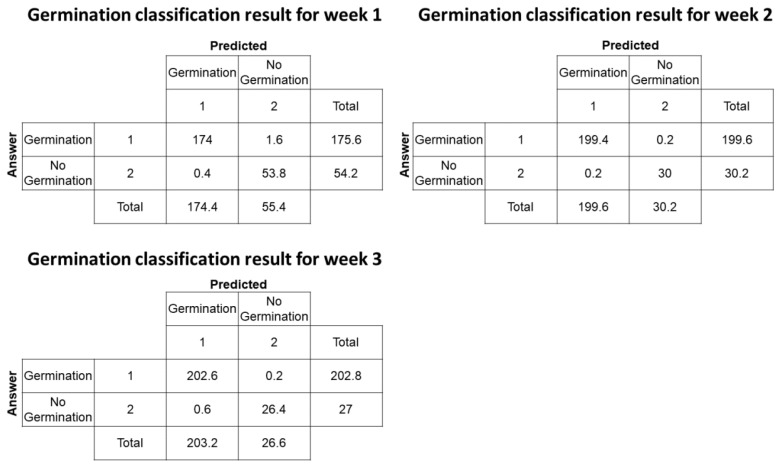
The representative confusion matrix for the germination classification model for Week 1, 2, and 3.

**Figure 2 plants-12-03867-f002:**
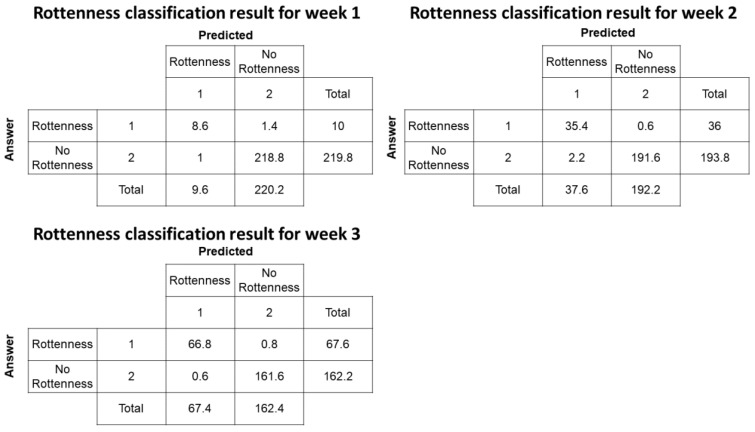
The representative confusion matrix for the rottenness classification model for Week 1, 2, and 3.

**Figure 3 plants-12-03867-f003:**
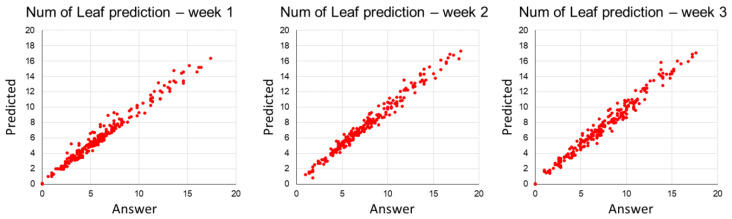
The regression result for number of leaves prediction from Week 1 to 3.

**Figure 4 plants-12-03867-f004:**
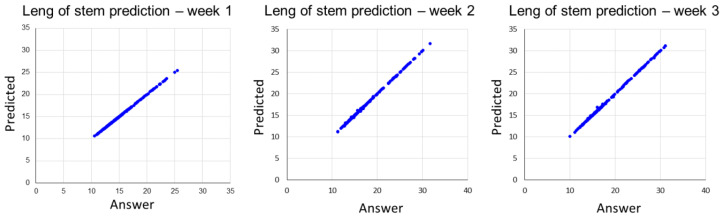
The regression result of predicting length of stem prediction from Week 1 to 3.

**Figure 5 plants-12-03867-f005:**
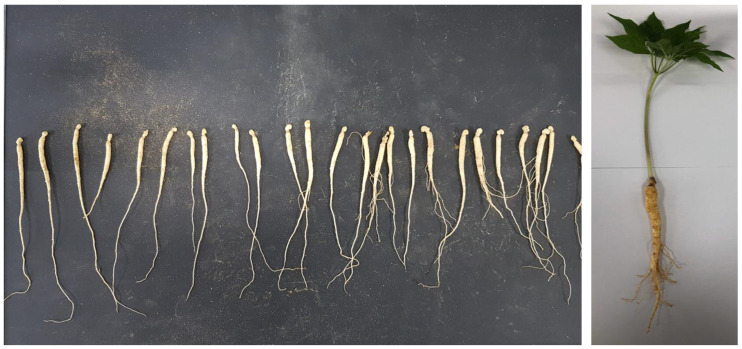
Representative image of Panax sprout ginseng 1-year-old seed ginseng.

**Figure 6 plants-12-03867-f006:**
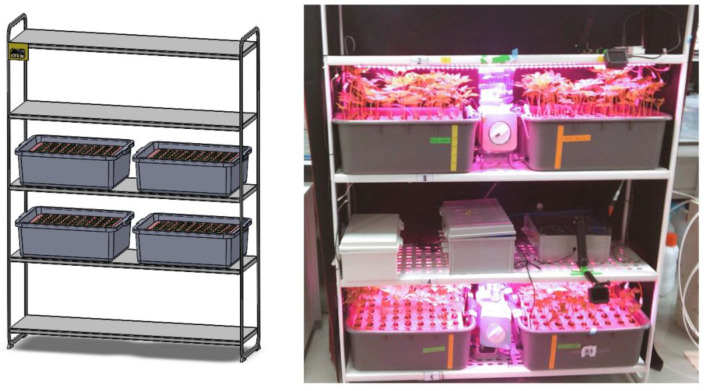
Hydroponic environment for growth of Panax sprout ginseng seed bed configuration.

**Figure 7 plants-12-03867-f007:**
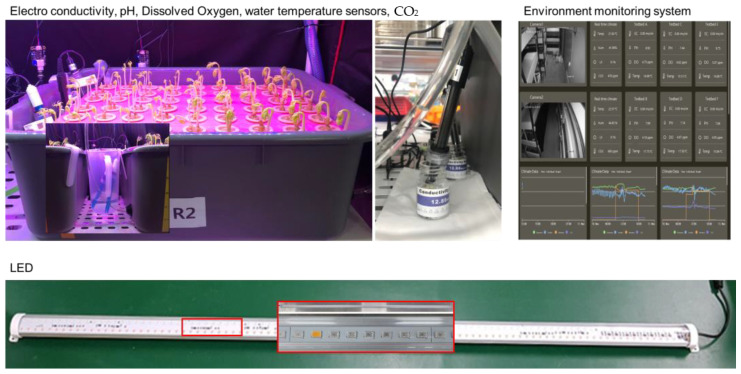
Various sensors applied to measure hydroponic environment condition, sensor monitoring system and LED.

**Figure 8 plants-12-03867-f008:**
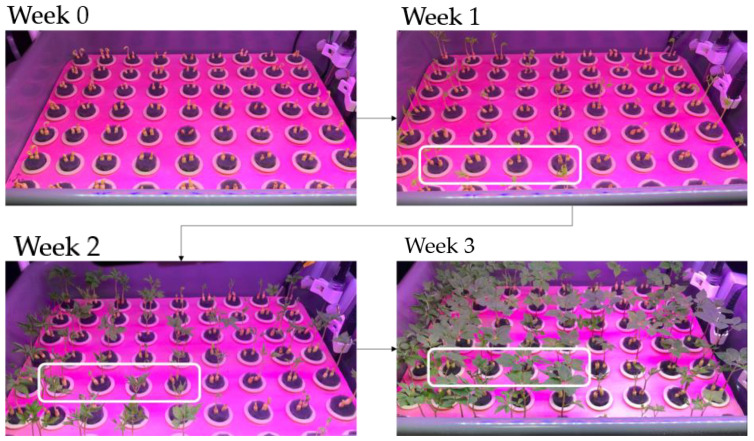
Representative growth of *Panax sprout* ginseng in on a hydroponic environment for 3 weeks.

**Figure 9 plants-12-03867-f009:**
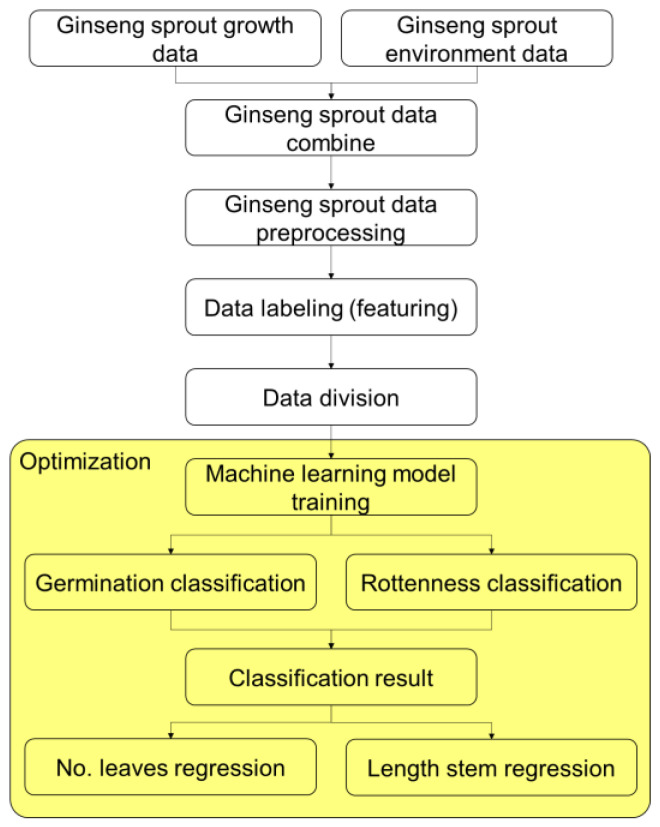
Overall flowchart of the proposed hierarchical machine learning-based *Panax sprout ginseng* growth classification and prediction model.

**Figure 10 plants-12-03867-f010:**
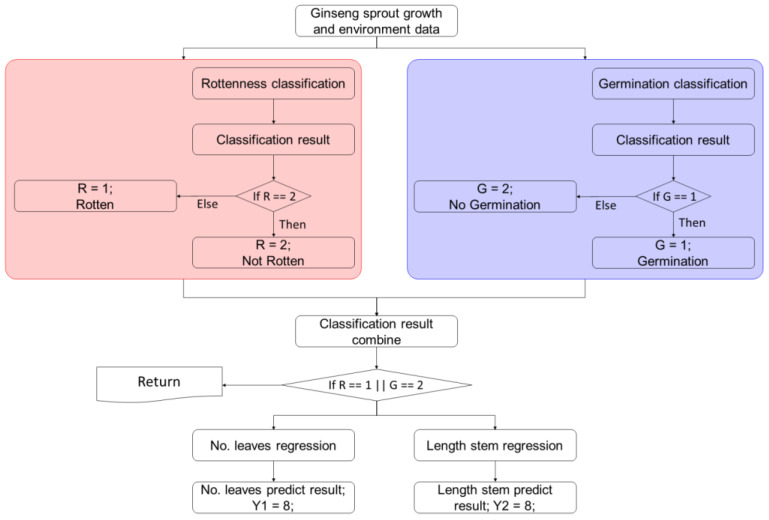
Detailed hierarchical machine learning-based classification and regression flowchart.

**Figure 11 plants-12-03867-f011:**
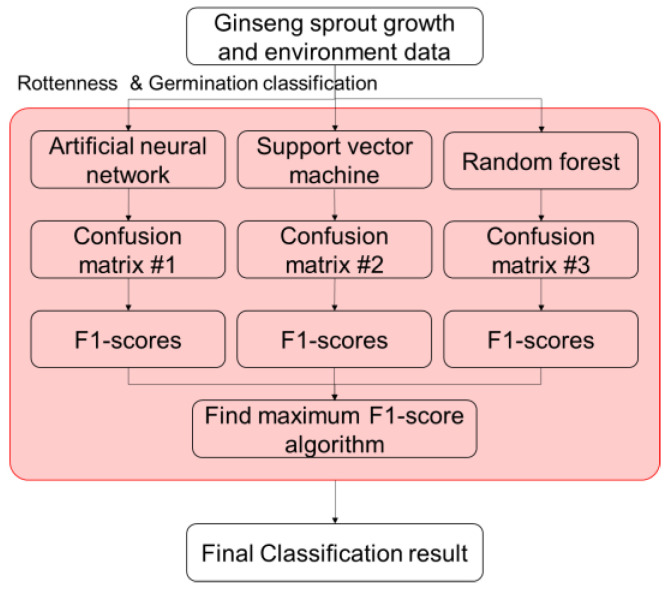
The *Panax ginseng* sprout rottenness classification using three different supervised machine learning algorithms.

**Table 1 plants-12-03867-t001:** Overall germination and rottenness classification performance for each week.

Germination Classification				
	Accuracy (%)	Precision (%)	Recall (%)	F1-Score (%)
Week 1	99.13 ± 0.28	99.77 ± 0.27	99.08 ± 0.61	99.42 ± 0.20
Week 2	99.82 ± 0.35	99.89 ± 0.20	99.89 ± 0.20	99.90 ± 0.20
Week 3	99.65 ± 0.51	99.69 ± 0.41	99.89 ± 0.21	99.79 ± 0.31

**Table 2 plants-12-03867-t002:** Overall rottenness classification performance for each week.

Rottenness Classification				
	Accuracy (%)	Precision (%)	Recall (%)	F1-Score (%)
Week 1	98.96 ± 0.52	87.16 ± 11.07	81.5 ± 19.21	83.47 ± 14.04
Week 2	98.78 ± 0.75	94.50 ± 3.56	98.31 ± 2.32	96.32 ± 2.10
Week 3	99.39 ± 0.58	99.12 ± 0.72	98.83 ± 1.43	98.97 ± 1.00

**Table 3 plants-12-03867-t003:** Overall performance of predicting the number of leaves and length of stem for each week from the proposed model is represented. nRMSE represents normalized root mean square error. MAE represents maximum absolute error. R represents the correlation coefficient value.

Number of Leaf Prediction			
	nRMSE	MAE	R
Week 1	0.27 ± 0.03	0.79 ± 0.07	0.96 ± 0.01
Week 2	0.20 ± 0.04	0.76 ± 0.16	0.98 ± 0.01
Week 3	0.18 ± 0.05	0.78 ± 0.23	0.98 ± 0.01

**Table 4 plants-12-03867-t004:** Overall performance of predicting the length of stem for each week from the proposed model is represented. nRMSE represents normalized root mean square error. MAE represents maximum absolute error. R represents the correlation coefficient value.

Length of Stem Prediction			
	nRMSE	MAE	R
Week 1	0.006 ± 0.004	0.014 ± 0.009	0.99 ± 0.00
Week 2	0.03 ± 0.002	0.05 ± 0.03	0.99 ± 0.00
Week 3	0.02 ± 0.01	0.04 ± 0.01	0.99 ± 0.00

**Table 5 plants-12-03867-t005:** Input feature of machine learning (ML) algorithms and output results from ML algorithm.

Sensor-Based Input Feature List		Growth Measured Input Feature List	
1	Average temperature	1	TGS nutrient solution type
2	Maximum temperature	2	Ginseng sprout year
3	Minimum temperature	3	Total length
4	Average water temperature	4	Ginseng sprout root length
5	Maximum water temperature	5	Ginseng sprout root diameter
6	Minimum water temperature	6	Ginseng sprout growth week
7	Average humidity	**1st stage ML result and 2nd stage ML input feature**	
8	Maximum humidity	1	Germination result
9	Minimum humidity	2	Rottenness result
10	Average pH	**2nd stage ML input feature**	
11	Maximum pH	1	Number of leaves
12	Minimum pH	2	Length of stem
13	Average EC		
14	Maximum EC		
15	Minimum EC		
16	Average DO		
17	Maximum DO		
18	Minimum DO		

## Data Availability

The data presented in this study are available upon request from the corresponding author.

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
