# Peer review of "Hierarchical Machine Learning-Based Growth Prediction Model of Panax ginseng Sprouts in a Hydroponic Environment"

_plants, 2023, doi:10.3390/plants12223867_

Round 1

Reviewer 1 Report

Comments and Suggestions for Authors The work is important to improve the production, not only of sprouts,
but in general for any crop, the work is interesting but needs to be
improved in different aspects that are described below:

Abstract

There is a lack of introduction with important aspects about the crop.

The results can be summarized a little, and the sentences can be shortened so that they are not so long in the abstract.

Keywords, change them to words that are not in the title.

Introduction

Panax ginseng is a scientific name, therefore in italics throughout the text.

L51-53, one citation is missing.

L56-58 missing citation.

Formulas and abbreviations such as CO2, should be in their correct form (CO2). Revise the entire text.

L68-70 citation of these studies is missing.

L76-78 Add quote on LEDs: Nájera, C.; Gallegos-Cedillo, V.M.; Ros, M.; Pascual, J.A. Role of Spectrum-Light on Productivity, and Plant Quality over Vertical Farming Systems: Bibliometric Analysis. Horticulturae 20239, 63. https://doi.org/10.3390/horticulturae9010063

L78-79 Add appointment

L83-85 Appointment

L85-87 Quote

L91-93 Quote

L105. Insert the symbol ºC instead of degrees.

L106. Add the full name to EC. Electrical conductivity (EC)

L131-134. Describe most important results with respect to Rottenness classification

Table 1. Are values in %? Specify

Figure 1. Describe further the results of the figure, I think it is important, but it is not given the importance.

L141-146. Describe the results in order of appearance of the tables and/or figures, otherwise the reader is lost.

Table 2. nRMSE, MAE, R even if the abbreviations are mentioned in the abstract, all abbreviations should be specified below the table.

L171. The same as above, the results should be described in order of appearance. Even describe first in a paragraph a table that later appears below.

L205-208 Citation

Discussion

More reinforced justification is needed, even if only by comparing with other models. More citations and more why of the results.

Materials and Methods.

I do not see the need for MMI images, Fig 7.

Conclusion

L436-437 No need to put in conclusions.

Describe further the advantages of using a model such as the one studied.

References

1-6 Does not correspond to bibliography, it should not be

Author Response

We greatly appreciate the reviewer’s hard work, which has improved the quality of our manuscript. We will provide a point-by-point response to the comments and suggestions as the reviewer indicated.

Reviewer 2 Report

Comments and Suggestions for Authors

The manuscript explains about a prediction technique for Panax Ginseng sprouts. However, the explanation requires significant improvement, although the target task may have some novelty. Moreover, the manuscript should be re-organized to provide a clear overview of the research.

1. Hierarchical machine learning-based prediction seems to lack novelty.

The authors suggested hierarchical approach for the given task. However, it seems to be just a sequential application of some machine learning models. The training process are not linked, so every model is independent. What can be differences between the suggested approach and a separated model training?

2. The ML methods are not described properly.

Since the model output was not used for physiological modeling, the only novelty of this study is machine learning approach. Therefore, the whole process such as data selection, data preprocessing for the input, model training, model evaluation, model selection, etc. must be carefully explained.

Author Response

(The authors gave the same response as above.)

Round 2

Reviewer 2 Report

Comments and Suggestions for Authors

Authors revised the paper properly.